

# A Fractal Framework for Channel-Hillslope Coupling

Benjamin Kargère[1,2], José Constantine[2], Tristram Hales[3], Stuart Grieve[4], and Stewart Johnson[1]

[1]Williams College Department of Mathematics
[2]Williams College Department of Geosciences
[3]Cardiff University, School of Earth and Environmental Sciences
[4]Queen Mary University of London, School of Geography

**Correspondence:** Benjamin Kargère (bkargere@gmail.com)

**Abstract.** Questions of landscape scale in coupled channel-hillslope landscape evolution have been a significant focus of geomorphological research for decades. Studies to date have suggested a characteristic landscape length that marks the shift from fluvial channels to hillslopes, limiting fluvial incision and setting the length of hillslopes. The representation of real-world landscapes in slope-area plots, however, makes it challenging to identify the exact transition from hillslopes to channels, owing to the existence of an intermediary colluvial valley region. Without a rigorous explanation for the scaling of the channel hillslope transition, the use of computational models, which are forced to implement a finite grid resolution, is limited by the scaling of the physical parameters of the model relative to the grid resolution. Grid resolution is also tied to the width of channels, which is undetermined without a rigorous explanation of where channels begin.

Building on existing work, we demonstrate the existence and implications of the characteristic landscape length and its relationship to grid resolution. We derive the characteristic landscape length as the horizontal length in a one-dimensional landscape evolution framework required to form an inflection point. On a two-dimensional domain, channel heads form in steady state at the characteristic area, the square of the characteristic length, independent of grid resolution. We present a box-counting fractal definition using the grid resolution, revealing that the dimension of the contributing drainage region on steady-state hillslopes is expressed as a multifractal system. In sum, channels have contributing drainage areas, therefore a dimension of two, whereas, by definition, unchannelized locations or nodes have a dimension between zero and two, so not a well-defined area. This conceptualization aligns with the observed scaling of channel width. It also importantly suggests that real-world landscapes have something analogous to the concept of a grid resolution, as this paper demonstrates. In doing so, our works clarifies several unresolved properties of channel-hillslope coupling, with potential for substantially improving the accuracy of coupled landscape evolution models in replicating landscape forms.

## 1 Introduction

Landscape evolution models (LEMs) are quantitative theories that describe the physical processes shaping geomorphic patterns. Coupled channel-hillslope LEMs combine both fluvial and hillslopes processes, each described by individual mathematical statements called geomorphic transport laws. A central, yet poorly understood aspect of these coupled LEMs is channel initiation, setting the transition between hillslopes and channels (Tucker and Hancock, 2010; Dietrich et al., 2003). In real-





world landscapes, this transition is often described in terms of stochastic perturbations and time-dependent behavior (Smith and Bretherton, 1972; Howard and Kerby, 1983; Del Vecchio et al., 2023). Given the prohibitive timescales of landscape evolution, the computational evaluation of LEMs is essential for elucidating landscape evolution over extended periods, further providing insights into the interconnected dynamics of hillslope and channel evolution (Tucker and Hancock, 2010).

Selecting a finite grid resolution is essential for both computationally evaluating two-dimensional coupled channel-hillslope

LEMs and interpreting real-world topography from digital elevation models (DEMs). Scaling grid resolution relative to physically derived parameters or topography presents several challenges. The widely-used stream-power incision model (Howard and Kerby, 1983; Howard, 1994; Whipple and Tucker, 1999) requires contributing drainage area as a proxy for discharge. When stream-power incision is combined with linear diffusion (Culling, 1960), the scaling of contributing drainage area across grid resolutions inconsistently affects hillslopes, with presumably parallel flow paths, and channels, where flow paths converge

(Pelletier, 2010; Hergarten, 2020; Hergarten and Pietrek, 2023; Bernard et al., 2022). Previous efforts have addressed this issue by proposing criteria for channel initiation (Hergarten, 2020; Hergarten and Pietrek, 2023), but the scaling relationships between the physically-derived model parameters and channel initiation remain unclear. Instead, computational models often implement a physically-derived or arbitrarily chosen threshold for drainage area or slope-area, below which stream-power erosion is absent (Perron et al., 2008; Tucker and Bras, 1998; Campforts et al., 2017). However, some studies have suggested

that the value of the channelization thresholds themselves depends on grid resolution (Montgomery and Dietrich, 1992; Ariza-Villaverde et al., 2015; Tarboton et al., 1991), and some studies have questioned the necessity of using a threshold altogether (Perron et al., 2008; Theodoratos et al., 2018). Grid resolution also complicates the modeling of channels with large drainage areas where the expected channel widths exceed the pixel width (Pelletier, 2010; Hergarten, 2020).

Prior research has also explored the characteristic landscape length that distinguishes hillslopes from channels and defines

the width of first-order valleys (Montgomery and Dietrich, 1992; Tarboton et al., 1988; Perron et al., 2008; Horton, 1945). This length is thought to be associated with the flow-path length from topographic maxima to inflection points with maximum steepness, differentiating convex summits and concave-up valleys (Willgoose et al., 1991; Tarboton et al., 1991; Roering et al., 2007). Additionally, research indicates that this length scale corresponds to the square root of the contributing area at channel heads (Montgomery and Dietrich, 1992; Tarboton et al., 1988; Perron et al., 2008; Tucker and Bras, 1998). Researchers working

on this problem have long noted three distinct regions—from hillslopes, intermediate colluvial valleys, and channels—in slope-area plots. In particular, they have been intrigued by the curved region associated with debris flows and shallow landslides, corresponding to unchannelized colluvial valleys with relatively constant slopes (Montgomery and Foufoula-Georgiou, 1993; Stock and Dietrich, 2006; Struble et al., 2023; McGuire et al., 2023). The connection between the intermediate region and debris flows is problematic, however, because curved slope-area plots appear in models and real-world landscapes without

debris flows.

In this work we analytically derive the characteristic landscape length from a one-dimensional analysis. We define a fractal box-counting definition using the characteristic landscape length and the pixel width as a measure. Unchannelized nodes, those with contributing drainage less than the characteristic length squared, do not have a well-defined contributing drainage area, instead varying with the grid resolution according to their fractal dimension. We demonstrate that this corresponds to both





computational simulations and real-world topography using Gabilan Mesa in California. Finally, we propose directions for computational models and suggest that real-world landscapes have a property analogous to a grid resolution.

## 2 Detachment-Limited, Linear Diffusion Landscape Evolution

Landscape evolution models for erosional drainage-basin evolution have a storied history. The simplest, most applicable model assumes detachment-limited stream-power (Howard, 1994; Whipple and Tucker, 1999) and linear diffusion (Culling, 1960).

$$\frac{\partial z}{\partial t} = U - K A^m |\nabla z|^n + D\nabla^2 z \qquad (1)$$

On a two-dimensional domain, topographic elevation, $z$, is a function of the horizontal coordinates $x$ and $y$, as well as time $t$. This model is based on three fundamental parameters: $K$, $D$, and $U$. Erodibility, $K$, modulates the strength of the stream-powered erosion. Diffusivity, $D$, characterizes the strength of gravity-driven erosive processes. The uplift rate, $U$, represents the effects of base-level forcing, such as uplift or base-level fall. We assume that $K$, $D$, and $U$ are constant in both space and
time.

Stream-power incision corresponds to the term $K A^m |\nabla z|^n$. $A$, the contributing area, a proxy for discharge in steady-state landscapes, is calculated for each $(x, y)$ node on the two-dimensional domain using flow-routing vectors in the direction of steepest descent of $z$, such that $A$ is a function of $x$, $y$, $z$, and $t$. $|\nabla z|$ is the norm of the gradient of $z$. Stream-powered erosion is assumed to be detachment-limited, meaning that sediments, once detached, are not redeposited within the domain. For Eq. (1), with contributing drainage area $A$, we set $m = \frac{1}{2}$, within the typical range of $0.4$ to $0.55$ observed for real-world topography
(Whipple and Tucker, 1999). Throughout this work we set $n = 1$, inducing linear behavior of stream-powered erosion and in the range suggested by empirical studies (Whipple and Tucker, 1999; Lague, 2014).

Linear diffusion (Culling, 1960), representative of mixing processes such as soil creep and bioturbation, assumes that the diffusive flux, $q_d$, is directly proportional to the gradient of $z$, given as $q_d = D|\nabla z|$. The divergence of the diffusive flux, $D\nabla^2 z$
can be positive or negative, indicating erosion in concave-down profiles and deposition in concave-up profiles. Linear diffusion accurately models soil-mantled landscapes with cohesive sediments and gradients significantly lower than the angle of repose (McKean et al., 1993). However, it fails to represent sediment fluxes accurately on hillslopes where local slopes approach a critical slope (Roering et al., 1999).

We define distinct height ($H$) and length ($L$) dimensions, in alignment with prior dimensional analyses (Theodoratos et al.,
2018). For steady-state topography, the horizontal dimension pertains to the two-dimensional domain, while the vertical dimension serves as the function's codomain, organized so that erosion balances uplift everywhere. In Eq. (1), $D$ has fundamental dimension $L^2 T^{-1}$, and $U$ has fundamental dimension $H T^{-1}$. The combination $m = \frac{1}{2}$ and $n = 1$ preserves the fundamental dimension of $K$ as a rate ($[K] = T^{-1}$).



## 2.1 One Horizontal Dimension

With two horizontal dimensions, contributing drainage area scales inconsistently with grid resolution owing to differences
in flow routing between hillslopes and channels. As noted by Pelletier (2010), Hergarten (2020), and Hergarten and Pietrek
(2023), hillslopes are thought to have parallel flow paths, whereas channels have convergent flow paths. On hillslopes, the
width of the parallel flow paths is a function of grid resolution, whereas regions with convergent flow are relatively unaffected
by changes in grid resolution. In order to address this inconsistency, we first consider the one-dimensional equivalent to the
landscape evolution model presented in Eq. (1):

$$\frac{dz}{dt} = U - K|x|\left|\frac{dz}{dx}\right| + D\frac{d^2z}{dx^2} \tag{2}$$

With the boundary length $\ell$:

$$-\frac{\ell}{2} < x < \frac{\ell}{2}$$

$|x|$, a length, is a proxy for the amount of the accumulated precipitation across the one-dimensional domain. The exponent
$m = 1$ preserves the fundamental dimension of $K$ as $T^{-1}$ in the one-dimensional framework. Bonetti et al. (2020) present
a non-dimensionalization of Eq. (2), using characteristic length $l_{\hat{c}} = \ell$, characteristic height $h_{\hat{c}} = \frac{U}{K}$, and characteristic time
$t_{\hat{c}} = \frac{1}{K}$. Therefore $\frac{x}{l_{\hat{c}}} = \hat{x}$, $\frac{z}{h_{\hat{c}}} = \hat{z}$, and $\frac{z}{t_{\hat{c}}} = \hat{t}$, resulting in:

$$\frac{d\hat{z}}{d\hat{t}} = 1 - |\hat{x}|\left|\frac{d\hat{z}}{d\hat{x}}\right| + (C_I)^{-1}\frac{d^2\hat{z}}{d\hat{x}^2} \tag{3}$$

$$-\frac{1}{2} < \hat{x} < \frac{1}{2}$$

$C_I$, the channelization index, is a Péclet number, explaining the competition between advection and diffusion (Perron et al.,
2008; Anand et al., 2023; Bonetti et al., 2020).

$$C_I = \frac{\ell^2 K}{D} \tag{4}$$





**Table 1.** Symbols and variable definitions used in the study

| Symbol | Dimension | Description |
|---|---|---|
| $x, y$ | $L$ | Horizontal Coordinates |
| $z$ | $H$ | Elevation |
| $t$ | $T$ | Time |
| $K$ | $\frac{1}{T}$ | Erodibility |
| $D$ | $\frac{L^2}{T}$ | Diffusivity |
| $U$ | $\frac{H}{T}$ | Uplift |
| $A$ | $L^2$ | Contributing Area |
| $\delta$ | $L$ | Pixel Width |
| $\ell$ | $L$ | Boundary Length |
| $r$ | $L$ | Linear-Diffusion Characteristic Length |
| $|\nabla z|$ | $\frac{H}{L}$ | Norm of Gradient of $z$ |
| $\nabla^2 z$ | $\frac{H}{L^2}$ | Laplacian of Elevation |
| $t_{\hat{c}}, t_{\tilde{c}} = \frac{1}{K}$ | T | Characteristic Times |
| $h_{\hat{c}}, h_{\tilde{c}} = \frac{U}{K}$ | H | Characteristic Heights |
| $\hat{t}, \tilde{t}, \hat{z}, \tilde{z}$, etc | 1 | Dimensionless Operators |

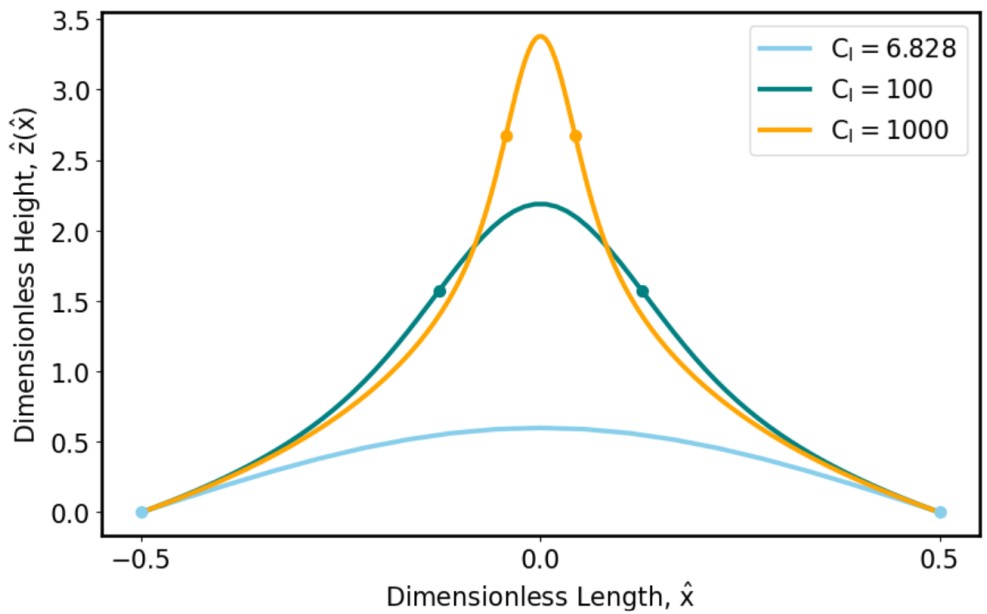

**Figure 1.** Equation 3 solved for various values of $C_I$. As noted in Anand et al. (2023), asymptotically large values of $C_I$, concave-down profiles are locally preserved near $\hat{x} = 0$ for asymptotically large values of $C_I$. Inflection points are plotted with dots. The value $C_I \approx 6.828$ sets the distance between the inflection points equal to the boundary length.





Figure 1 plots the numerically evaluated steady-state profiles of Eq. (3) for various values of $C_I$. Larger values of $C_I$ manifest in narrower hillslope profiles relative to the boundary size 3, formed by the relative strength of advective processes over diffusive processes. The computational granularity necessary for accurate numerical evaluation is a function of the channelization index, $C_I$. Larger values of $C_I$ require a finer resolution to accurately represent the concave-down region around $\hat{x} = 0$.

As noted by Litwin et al. (2022a), the channelization index ($C_I$) is a dimensionless boundary length. The intrinsic length of Eq. (2) corresponds to a group involving $D$, since $[D] = L^2 T^{-1}$. To solve for this intrinsic length, we consider the fixed points of Eq. (3) (Howard, 1994). The fixed point of $\hat{z}$, where $\frac{d\hat{z}}{d\hat{x}} = 0$, is located at the top of the ridge at $\hat{x} = 0$. For $\hat{x} = 0$, stream-powered erosion does not occur, so the steady-state dimensionless erosion at the top of the ridge is $C_I^{-1} \cdot \frac{d^2 \hat{z}}{d\hat{x}^2} = -1$ (Roering et al., 2007). Inflection points, fixed points of $\frac{d\hat{z}}{d\hat{x}}$, occur at the sides of the ridge where $|\hat{x}| \left| \frac{d\hat{z}}{d\hat{x}} \right| = 1$. These inflection points mark the transition from concave-down to concave-up profiles. We denote $r$ as the length between $x = 0$ and the inflection points of Eq. (2), identical to the characteristic hillslope length given in prior works (Roering et al., 2007; Perron et al., 2008; Willgoose et al., 1991). For $C_I \approx 4 + 2\sqrt{2} \approx 6.828$, the inflection points of $\frac{dz}{dx}$ occur at $\pm \frac{\ell}{2}$, thus $\ell = 2r$. Solving for $r$:

$$r = \sqrt{\frac{D}{K} \left( \frac{\sqrt{2}}{2} + 1 \right)} \propto \sqrt{\frac{D}{K}} \tag{5}$$

For the remainder of this paper, we adopt $r$ as the characteristic landscape length, proportional to the characteristic length-scale $\sqrt{\frac{D}{K}}$ used in several prior studies (Perron et al., 2008; Theodoratos et al., 2018). $r$ can also be solved for from Eq. (2) as a first-order ordinary differential equation in $dz/dx$ (Appendix A).

## 2.2 Two Horizontal Dimensions

On a two-dimensional domain, stream-power linear diffusion landscape evolution is dependent on three horizontal lengths: the landscape length $r$, the boundary length $\ell$, and the pixel width $\delta$. The explicit dependence on $\delta$ is controversial. On a two-dimensional domain, $\delta$ is not only correlated with numerical error, as on a one-dimensional domain, but also sets the width of linear elements, such as channels. Considering these three lengths—the boundary length, the pixel width, and $r$—a minimum of two dimensionless groups can be formed, without unnecessarily rescaling the other dimensional quantities (Buckingham, 1914). This approach maintains the characteristic height as $\frac{U}{K}$, diverging from the approach of Litwin et al. (2022b).

Bonetti et al. (2018) define specific area, $a$, a length, as the contributing area per unit contour width in the limit approaching zero contour width (Bonetti et al., 2018; Gallant and Hutchinson, 2011). Specific area, satisfying the conservation of a unitary precipitation rate, is expressed as:

$$-\nabla \cdot \left( \frac{a \nabla z}{|\nabla z|} \right) = 1 \tag{6}$$

Given $\delta$, an infinitesimally small contour width cannot be achieved, and thus $a$ is not well-defined. In the following analysis, the specific drainage area $a$ is defined as $a = \frac{A}{\delta}$, while noting that the specific drainage area is implicitly dependent on the grid resolution (Litwin et al., 2022b; Costa-Cabral and Burges, 1994). Using the D8 flow routing algorithm on a square pixel





(Tarboton et al., 1988), both the flow length and contour width between diagonal pixels measure $\sqrt{2}\delta$ diagonally, whereas along the cardinal directions, both metrics equal $\delta$. With the $D\infty$ algorithm on a square pixel, the contour width and the flow length between neighboring pixels is always $\delta$ (Gallant and Hutchinson, 2011). Throughout this paper we use the $D\infty$ algorithm to bypass complexities in cardinal and diagonal directions and to accurately model flow-routing on divergent hillslopes.

The exponent $m = \frac{1}{2}$, applied to contributing area $A$ in Eq. (1), is commonplace for two reasons. First, assuming $n = 1$, the
scaling of channel slopes with the square root of drainage area is consistently observed in bedrock channels and large drainage basins (Leopold and Maddock Jr, 1953). Additionally, the choice $m = \frac{1}{2}, n = 1$, preserves the fundamental dimension of $K$ as a rate. The landscape evolution models in the form of Eq. (1) with $m = \frac{1}{2}$ do not explicitly address the role of flow width (Perron et al., 2008). Previous attempts to account for the channel width have sought to normalize the fluvial erosion component by a factor of $\frac{w}{\delta}$, where $w$ is the channel width (Howard, 1994; Perron et al., 2008). Channel widths are observed to scale with the
square root of drainage area Leopold and Maddock Jr (1953), thus $\frac{w}{\delta} \propto \frac{\sqrt{A}}{\delta}$.

$$\frac{\partial z}{\partial t} = U - \frac{\sqrt{A}}{\delta} K \sqrt{A} |\nabla z| + D\nabla^2 z \tag{7}$$

$$= U - Ka|\nabla z| + D\nabla^2 z \tag{8}$$

Rescaling the stream-powered erosion by $\frac{\sqrt{A}}{\delta}$ defines the specific drainage area, $a$, a length, as the discharge source term, preserving the dimension of $K$. This normalization implies that fluvial erosion is proportional to contributing area, and thus
discharge, assuming a uniform precipitation rate. The calculation of $a$ with contour width $\delta$ is appropriate for unchannelized nodes, such that erosion occurs by overland flow and therefore the minimum contour width is $\delta$. We first explore the implications of Eq. (8) for unchannelized nodes. The use of $a$ ensures that the contributing drainage of parallel locally parallel on hillslopes are independent of pixel width, thereby conforming to a one-dimensional framework. We will show that parallel flows occur locally for at the inflection point, $a = r$. For nodes without $a = r$, the specific drainage area is implicitly dependent
on $\delta$. This dependence is unavoidable but can be reconciled through an additional dimensionless group.

### 2.3 Dimensional Analysis

As previously noted, considering the three horizontal lengths $\ell$, $\delta$, and $r$ entails at least two dimensionless groups. Non-dimensionalizing Eq. (8) with $l_{\tilde{c}} = r$, $h_{\tilde{c}} = \frac{U}{K}$, $t_{\tilde{c}} = \frac{1}{K}$, such that $\frac{a}{l_{\tilde{c}}} = \tilde{a}$, $\frac{x}{l_{\tilde{c}}} = \tilde{x}$, $\frac{y}{l_{\tilde{c}}} = \tilde{y}$, $\frac{z}{h_{\tilde{c}}} = \tilde{z}$, and $\frac{t}{t_{\tilde{c}}} = \tilde{t}$ results in:

$$\frac{\partial \tilde{z}}{\partial \tilde{t}} = 1 - \frac{r}{\delta}\tilde{A}\left|\tilde{\nabla}\tilde{z}\right| + \frac{D}{r^2 K}\tilde{\nabla}^2 \tilde{z} \tag{9}$$

$$= 1 - \tilde{a}\left|\tilde{\nabla}\tilde{z}\right| + \left(\frac{\sqrt{2}}{2} + 1\right)^{-1}\tilde{\nabla}^2 \tilde{z} \tag{10}$$

with boundary length:

$$\frac{\ell}{l_{\tilde{c}}} = \frac{\ell}{r} = \ell \cdot \sqrt{\frac{K}{D(\frac{\sqrt{2}}{2} + 1)}} \propto \sqrt{(C_I)}$$



Therefore, by considering the form with the dimensionless area, this yields two dimensionless groups related to the horizontal length scales:

$$\Pi_1 = \frac{r}{\delta}$$

With the second appearing in the boundary condition:

$$\Pi_2 = \ell \cdot \sqrt{\frac{K}{D(\frac{\sqrt{2}}{2} + 1)}}$$

The dimensionless group $\frac{r}{\delta}$ represents the number of pixels of contributing drainage, though not necessarily individual pixels (with multiple flow-direction routing algorithms), required to form an inflection point. Identical values of this dimensionless group create a consistent scaling break in steady-state slope-area space. With increased resolution, the inflection point occurs for fewer pixels of contributing area, but the same specific drainage area calculated using contour width $\delta$. The dimensionless ratio $\frac{\ell}{r}$ denotes the boundary relative to $r$, and is proportional to the square root of the channelization index. These dimensionless groups function similarly to two controls of a microscope: one control alters the proximity of the microscope to the specimen, effectively changing the boundary size, and another that adjusts the resolution through the focus of the lenses. The effect of these groups is demonstrated through computational simulations in the following section.

## 3   2-Dimensional Results

We completed steady-state numerical simulations using Anand et al. (2023)'s Landlab-equipped (Hobley 2017) Jupyter note-book, using D∞ algorithm (Anand et al., 2020) and implicit diffusion. Our simulations follow Eq. (8), with specific drainage area calculated using $\delta$ as a contour width.

Inflection points occur at $a = r$ (Fig. 2). The specific drainage area value $a = \frac{r^2}{\delta}$, features local fluvial power-law scaling according to $-\frac{m}{n} = -\frac{1}{2}$, with slope decreasing steadily with the square root of specific drainage area. For specific drainage area much greater than $a = \frac{r^2}{\delta}$, the gradient scales with the specific drainage area according to $-\frac{m}{n} = -1$. The significance of the specific drainage area $a = \frac{r^2}{\delta}$ is explained in the following section.

Larger values of $\frac{\ell}{r}$ induce larger boundary sizes relative to $r$, therefore portraying more hillslopes and larger stream orders (Fig. 3). The dimensionless group $\frac{r}{\delta}$ controls the magnification (Fig. 4). Nodes with drainage area exceeding $A = r^2$ are shown with yellow dots. For different values of $\delta$, the resolution changes, but the images stay largely the same.




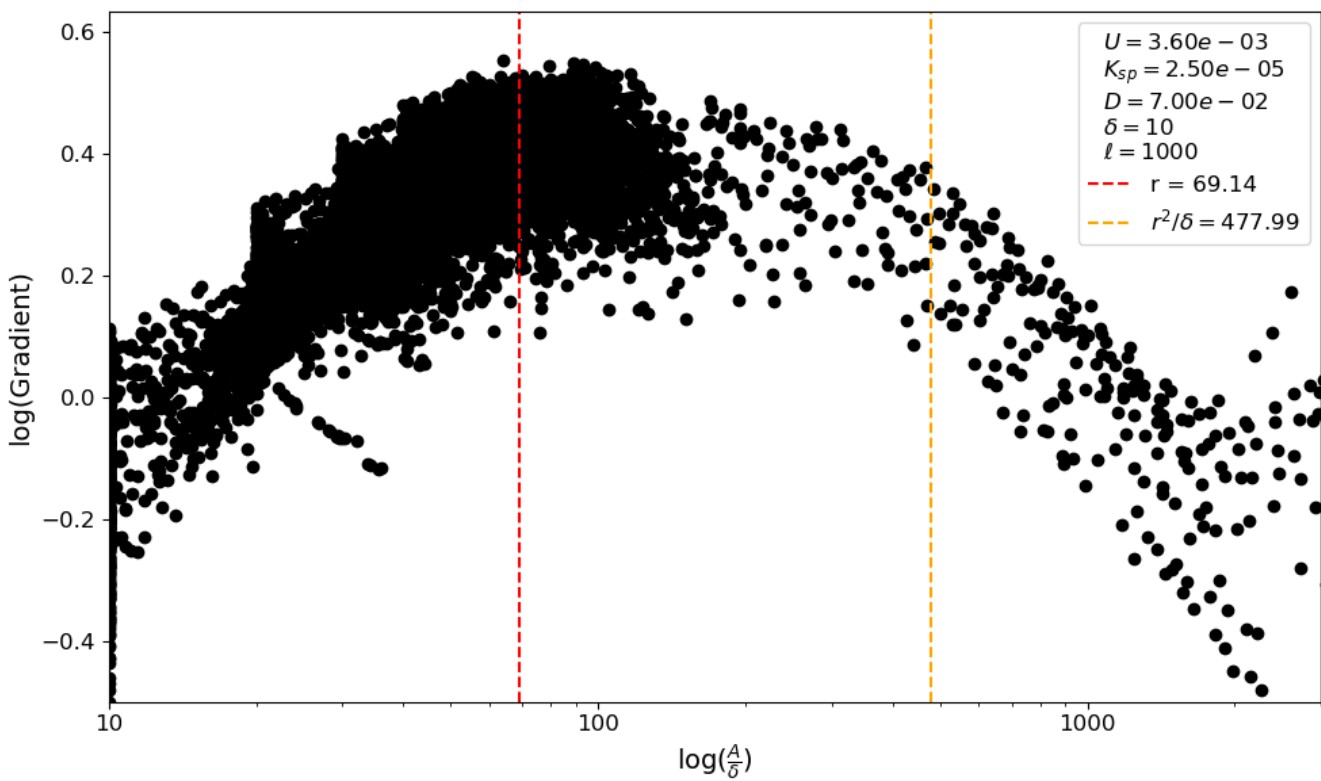

**Figure 2.** Slope-specific drainage area plot for the simulated topography shown in Fig. 3 for $\ell = 1000$. The specific drainage area $a = r$, shown with the red line, has the steepest slope. The specific drainage area value $\frac{r^2}{\delta}$, shown in orange, represents the transition to fluvial power-law scaling, with slope decreasing steadily with contributing area. This value is resolution independent as an area, but not as a length.



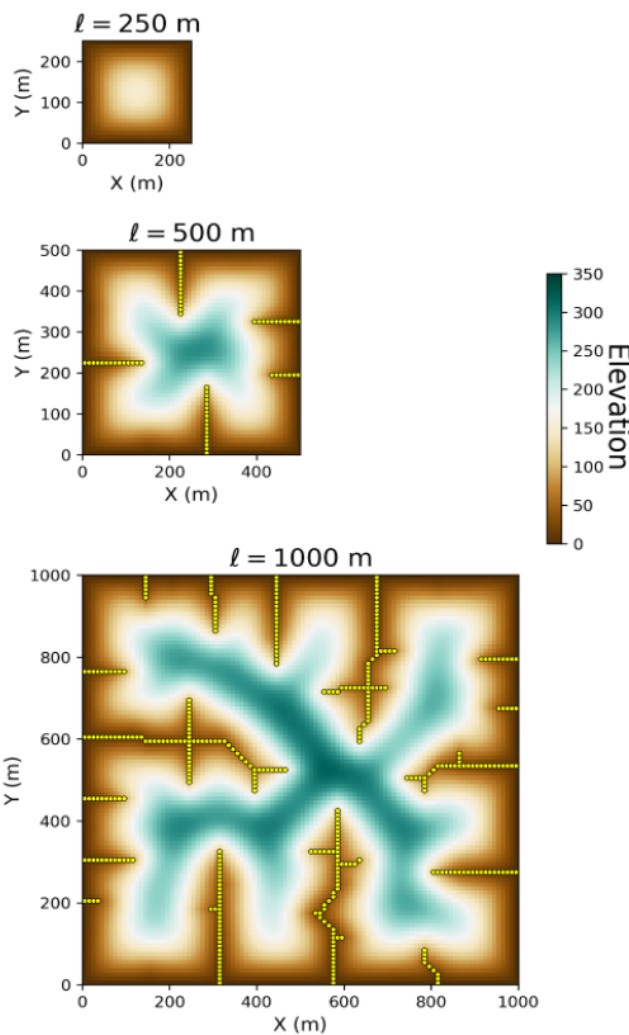

**Figure 3.** Computational simulation results varying $\ell$ for $\delta = 10$, $K = 2.5e-5yr^{-1}$, $D = 7e-2m^2yr^{-1}$, $U = 3.6e-3myr^-1$. The corresponding channelization indices, $CI = \frac{l^2 K}{D}$, are approximately 22, 89, and 357. These simulations preserve the dimensionless group $\frac{r}{\delta}$, the ratio of the characteristic length to the pixel width. Larger values of $C_I$, given the same $r$ value, imply a resized boundary length, $\ell$. Channels form for $A = r^2$, shown with yellow dots.





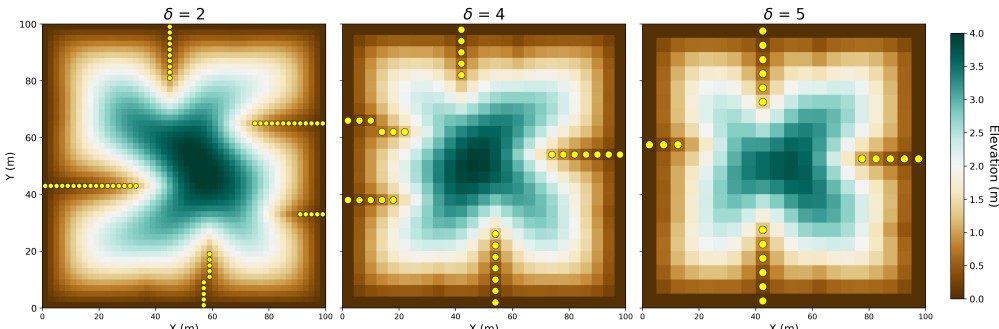

**Figure 4.** Computational results varying $\delta$ for $U = 1e-3myr^{-1}$, $D = 5e-2m^2yr^-1$, $K = 2.5e-4yr^{-1}$. Nodes with drainage area exceeding $a = \frac{r^2}{\delta}$ are shown with yellow dots. Varying $\delta$ with $\frac{r}{\ell}$ constant produces similar results. With decreasing $\frac{r}{\delta}$, maximum elevation decreases, since topographic maxima have specific drainage area $\delta$. This presents a possible benefit for subtracting $\delta$ from $a$, thus counting the number of links (Rodríguez-Iturbe et al., 1997). We forgo this approach for simplicity.



## 4 Fractal Analysis

In the steady-state simulations presented in Fig. 3 and Fig. 4, channel heads are located at $a \approx \frac{r^2}{\delta}$. This value is well-defined

as an area, $r^2$, as hypothesized by Montgomery and Foufoula-Georgiou (1993), but not as a $\delta$-derived specific drainage area.

On a discrete grid with pixel width $\delta$, points, with dimension zero, have an area of $\delta^2$, formed by a length and width of $\delta$. Topographic maxima have zero-dimensional contributing drainage area, $\delta^2$. Lines, with no width, have width $\delta$. Resolution-independent lengths on steady-state topography correspond to the inflection points, $a = r$ for linear diffusion. Using the $D\infty$ algorithm, these lengths correspond to flow routing lengths, rather than Euclidean lengths. The flow directions for nodes

with $a = r$ are locally parallel in $\lim_{\delta \to 0}$, separating zones of topographic divergence ($a < r$), from topographic convergence ($a > r$). Areas on two-dimensional grids are resolution independent. Numerical simulations (Fig. 3 and Fig. 4) indicate that these contributing areas correspond to nodes with $A = r^2$, such that the contributing area is convergent to a point, i.e. a channel head. In these simulations according to Eq. (8), channels, linear elements which by definition have width $\delta$, form downstream from these channel heads.

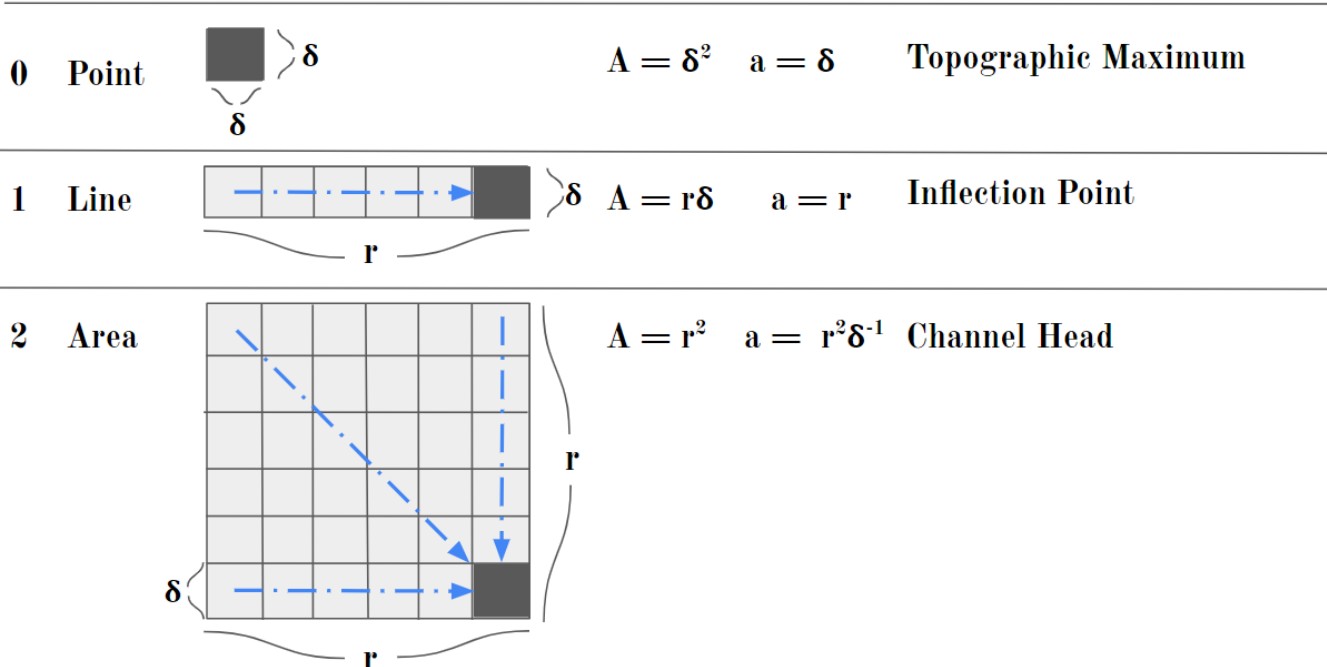

**Figure 5.** Schematic diagram of the non-fractal dimension drainage regions. On landscapes, lengths do not correspond to Euclidean lengths, but instead flow lengths. Areas are not necessarily square-shaped.



## 4.1 Fractal Definition

The resolution-independent dimension of unchannelized nodes, with $a \leq \frac{r^2}{\delta}$, is not necessarily an integer. Nodes in locations with flow directions of partial convergence or partial divergence have non-integer, or fractal dimension. The dimension of unchannelized nodes with integer dimension, using $\delta$ as a contour width, have:

$$a = \frac{(r)^{D_f}}{\delta^{D_f - 1}} \tag{11}$$

$D_f$ is the box-counting fractal dimension, commonly defined as:

$$D_f = \frac{\log(\text{number of self-similar pieces})}{\log(\text{magnification factor})}$$

Solving for $D_f$,

$$D_f = \frac{\log(\frac{a}{\delta})}{\log(\frac{r}{\delta})} \tag{12}$$

The numerator $\log(\frac{a}{\delta})$ is a number of contributing pixels. The denominator, $\log(\frac{r}{\delta})$ is the magnification factor, one of the dimensionless groups, referring to number of contributing pixels at the inflection point. This fractal dimension is similar to definitions of Péclet numbers given in previous studies (Theodoratos et al., 2018; Hooshyar et al., 2020). Figure 6 shows the fractal dimension of the contributing drainage plotted for the parameters given in Fig. 2.

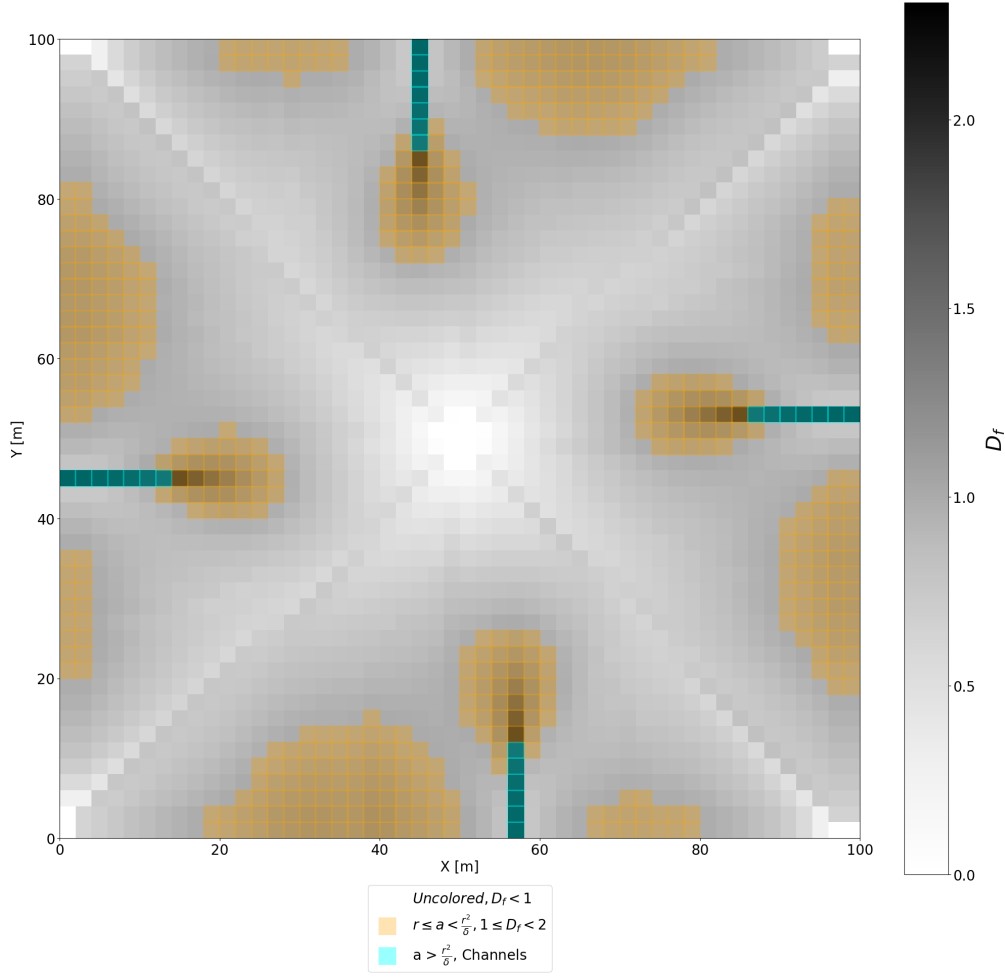

**Figure 6.** Simulation result for $U = 1e-3myr^{-1}$, $D = 5e-2m^2yr^-1$, $K = 2.5e-4yr^{-1}$, $\delta = 2, \ell = 100$ plotted according to the fractal dimension, $D_f$ (Eq. 12). Channel heads, with drainage area $r^2$, have dimension two. Colluvial valleys correspond to $1 < D_f < 2$. River segments have dimension two when accounting for channel width.



Equation (12) indicates that nodes with locally divergent flow near hilltops exhibit a contributing drainage region with fractal dimension between 0 and 1. Unchannelized nodes with locally convergent flow have a contributing drainage region with fractal dimension between 1 and 2, corresponding to unchannelized valleys (Fig. 6). This explanation clarifies the properties
of unchannelized nodes but does not extend to channelized nodes.

Equation (12) suggests that for channelized nodes, with $a \geq \frac{r^2}{\delta}$, $D_f$ exceeds two for $a$ calculated using $\delta$ as a contour width. Several studies have demonstrated that the fractal dimension of channel networks, as measured by their length-to-bifurcation ratios, approaches two for large stream orders (Tarboton et al., 1988; Rinaldo et al., 1998; La Barbera and Rosso, 1989; Rodríguez-Iturbe et al., 1997). While these fractal definitions may be independent, it would be unconventional for the box-
counting fractal dimension of a region on a two-dimensional surface to exceed two. For river segments observed on nature, channel width scales with the square root of drainage area (Leopold and Maddock Jr, 1953). Letting $\delta$ be the channel width at the channel head:

$$w = \frac{\delta}{r}\sqrt{A} \tag{13}$$

Box-counting with $\delta$-sized pixels for river nodes downstream of the channel head captures only the contributing nodes for a
fraction of the contour width of the channel. Using $w$ as the contour width for river nodes, $A = a_w \cdot w$, where $a_w$ is the specific contributing area corresponding to contour width $w$. Substituting $a_w$ for $a$ and $w$ for $\delta$ in the numerator of $D_f$, as defined for hillslopes, $D_f = 2$ for all channels. The ratio $\frac{r}{\delta}$ remains constant, the dimensionless magnification factor set by the parameters of the model.

$$w = \left(\frac{\delta}{r}\right)^2 \cdot a_w \qquad D_f = \frac{\log\left(\frac{a_w}{w}\right)}{\log\left(\frac{r}{\delta}\right)} \tag{14}$$

## 5 Confirmation and Discussion

Gabilan Mesa is a frequent subject for studies of linear-diffusion advection landscape evolution, featuring soil mantled hillslopes with cohesive soils and uniform steady-state topography. Debris flows and shallow landsliding are uncommon. Gabilan Mesa also has low gradients in most places relative to the critical slope, supporting the assumption of linear diffusion (Perron et al., 2008, 2009). Our data originate from a 2015 DEM, as utilized by (Grieve et al., 2016). These data include channel
heads using the Pelletier algorithm (Pelletier, 2013), as implemented by the LSDTopoTools (Pelletier, 2013), which estimates channel head locations from contour curvature. Floodplains were manually mapped and excluded. We did not need to run a sink-filling algorithm, since Gabilan features very little topographic roughness.

Figure 7 plots the gradient by drainage area for the section of the DEM shown in Fig. 8. The characteristic length corresponding to the inflection point is 62m, the drainage area of which varies by a factor of $\delta$ across resolutions (Bernard et al.,
2022). The drainage area of channel heads is resolution-independent. Figure 8 confirms that this corresponds to the location of channel heads as identified by the contour curvature method of Pelletier (2013). While $D$ can be consistently defined by hilltop





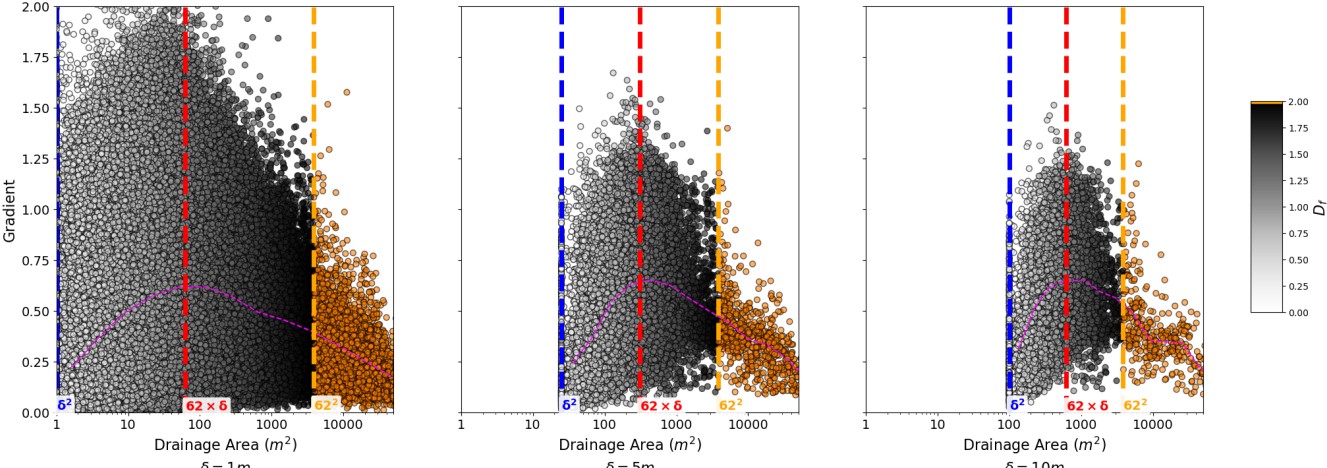

**Figure 7.** Log-log plot of slope by drainage area for nodes in Fig. 8, corresponding to 1, 5, and 10 meter pixels. The binned median is plotted in a magenta dashed line. The characteristic length is 62m. The minimum drainage area, corresponding to topographic maxima is $\delta^2$. The drainage area of the inflection point is $62\delta m^2$. Channel heads have the characteristic area of contributing drainage area which is resolution-independent.

curvature, $K$ is a relatively free parameter. There are several approaches for determining $K$, such as by drainage density or by channel steepness as $k_{sn} = (U/K)^{-m/n}$. However, this would assume that erodibility is constant throughout the domain (both hillslopes and channel networks). For this reason, as well as the potential for weakly nonlinear diffusion, we refrain
from defining the characteristic length as $r$ in this case-study. Nonetheless, the coherence between the topographic analyses presented in Fig. 7 and Fig. 8 and the mathematical framework derived from computational results confirms the utility of our theory (Bras et al., 2003).





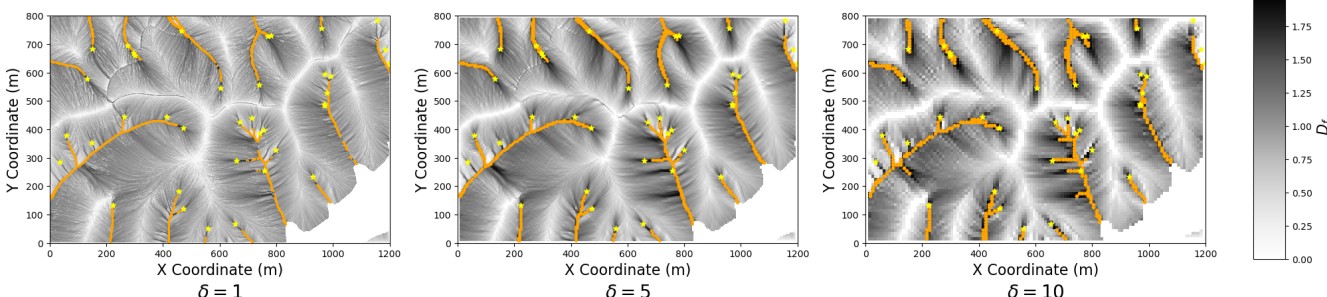

**Figure 8.** A section of Gabilan Mesa visualized by plotting the fractal dimension, $D_f$, for each node. Subsequent plots were generated by resampling the topographic elevation from the original 1m DEM to 5m and 10m resolution. The D-infinity algorithm calculated flow accumulation for every pixel. The fractal dimension is calculated according to Eq. (12), using the characteristic length of 62 m, derived from Fig. 7. Channelized pixels, those with $A > (62m)^2$, and thus a fractal dimension of greater than two, are highlighted in orange. This algorithm is resolution independent up to greater numerical error on coarser resolutions. Pelletier-algorithm (Pelletier, 2013) derived channel heads are marked with yellow stars, demonstrating a surprising correspondence between the channel head methods. Nodes with a fractal dimension of one separate regions of topographic divergence from topographic convergence.

## 5.1  Spatially Resolving Channel Width

The numerical simulations shown in this paper correspond to Eq. (8), with specific area calculated with contour width $\delta$. These
simulations have channel slopes that scale according to a concavity of $\frac{m}{n} = 1$, rather than a concavity of $\frac{m}{n} = \frac{1}{2}$, because the scaling in channel width is confined to a single pixel. Previous approaches, as given by Pelletier (2010), assume that pixel widths exceed channel widths throughout the simulated domain, such that pixel elevations represent the channel elevation within the pixel, rather than the average elevation. This results in the form $\frac{A}{w}$ for channels, producing the correct scaling for channel slope but forgoing the spatial representation of widening channels. Rather than assuming that pixel widths exceed
channel widths throughout the domain, we propose instead that models should assume that channel widths exceed the pixel width, with the pixel width equal to the channel head width. With this assumption, flow partitioning across multiple pixels could accommodate widening channels (Gailleton et al., 2024), enabling the precise depiction of both river longitudinal profiles and widths.

## 6  Arbitrary $\frac{r}{\delta}$

For large $\delta$ relative to $r$, landscapes are largely self-similar (Anand et al., 2023; Hergarten, 2020), with concave-down hill-slope profiles restricted to single pixels. These simulations converge to purely fluvial topography for $\frac{r}{\delta} \to 0$, with channel scaling-behavior occurring instantly. Likewise, previous works considered $\delta$ as an arbitrary value, seeking relationships to vary between $\delta$ (Pelletier, 2010). We suggest no such abstraction to vary between $\delta$ for numerical simulations. Real-world landscapes typically feature $r \gg \delta$, introducing computational challenges that can be mitigated by choosing a small $\frac{\ell}{r}$.




## 6.1 Physical Merit to $\delta$

In computational models, the channel head, with upstream contributing drainage $r^2$, marks the beginning of channels, linear elements of width $\delta$. In $\lim_{\delta \to 0}$, channel heads are points, resulting in precisely linear channels regardless of the area scaling. This is not the behavior observed in nature. In the $\lim_{\delta \to 0}$, the topography of steady-state computational simulations minimizes numerical error to form a smooth surface, whereas real-world landscapes feature roughness at increasingly small scales (Roering, 2008). This indicates the importance of other length scales in real-world landscapes, including the particle size (Andrle and Abrahams, 1989; Sangireddy et al., 2017; Sweeney et al., 2015; Dunne and Jerolmack, 2020), which are omitted from one-dimensional analyses. Studies have shown that showed the particle size distribution is closely related to the transition from slope-invariant colluvial valleys (i.e. $1 < D_f < 2$) and fluvial channels, suggesting a maximum particle size at the channel head (Neely and DiBiase, 2023).

## 6.2 Nonlinear Diffusion

Physical landscapes are influenced by a variety of processes not explicitly considered in this analysis, such as soil production (Heimsath et al., 1997), nonlinear diffusion (Roering et al., 1999), and groundwater infiltration (Litwin et al., 2022b). The fractal dimension of drainage region is a result of convergence and divergence on topography, and is not inherent to the one-dimensional equation with linear diffusion. Future work should seek to generalize this work to a variety of flux laws, such as nonlinear diffusion law in the Andrews-Bucknam form (Roering et al., 1999; Andrews and Bucknam, 1987), depth-dependent nonlinear diffusion (Roering, 2008), and nonlocal models (Foufoula-Georgiou et al., 2010; Furbish and Roering, 2013). For nonlinear diffusion in the form of Roering et al. (1999) and Andrews and Bucknam (1987), the characteristic length is not proportional to $\sqrt{\frac{D}{K}}$, but also a function of uplift relative to the critical slope. Identifying the characteristic length from topography with strongly nonlinear diffusion is inherently challenging, as hillslopes nearing the critical slope become planar (Roering et al., 2007). Additionally, computational models incorporating nonlinear diffusion must address the effect of the grid resolution setting the spatial scale of the diffusion process (Ganti et al., 2012; Furbish and Roering, 2013).

## 6.3 Curved Slope-Area Plots

The reasons for curved-slope area plots have long been a topic of discussion in geomorphology (Montgomery and Foufoula-Georgiou, 1993). Colluvial valleys, corresponding to curved region between hillslopes and channels in slope-area plots have often been attributed to the role of stochastic processes, such as debris flows and shallow landslides. We showed that for these regions, drainage area is not a reliable metric. For analyses of natural landscapes, our theory should help elucidate the role of debris flows and landslides (McGuire et al., 2023). For computational simulations, this intuition shows potential for testing theories related to the frequency of forcing, both tectonic and climatic, in hillslope-channel coupled landscape evolution (Godard and Tucker, 2021).



## 7   Conclusions

We showed that the relationship between the characteristic landscape length and grid resolution a stream-power plus linear-diffusion landscape evolution model is expressed as a multifractal system for unchannelized nodes. Nodes with locally divergent flow have a contributing drainage region with fractal dimension between 0 and 1, while unchannelized nodes with locally convergent flow display dimensions between 1 and 2, aligning with unchannelized valleys. Channels have well-defined contributing area, aligning with the observed scaling of channel width and channel slopes as the square root of drainage area. This finding underscores a significant parallel between computational grid resolution and real-world landscape features, the channel head and particle width, in particular. This study serves as a foundational step towards understanding geomorphic channel-hillslope coupling, highlighting the coherence and limitations of one-dimensional and two-dimensional landscape evolution equations.

*Code availability.* Jupyter notebooks for generating the figures are provided in a Zenodo repository.

## Appendix A:  Alternate Derivation of r

Let $y = \frac{dz}{dx}$, $y' = \frac{d^2 z}{dx^2}$. Let $x \geq 0, y \leq 0$, by symmetry. Then rearranging Eq. (2), we have the form of a first-order linear ODE in $y$, which can be solved by the integrating factor method.

$$y' + \frac{K}{D}xy = -\frac{U}{D}$$

$$\mu(x) = e^{\frac{K}{2D}x^2}$$

$$y = -\frac{U}{D}e^{-Kx^2/2D}\int\limits_0^x e^{\frac{K}{2D}\bar{x}^2}\, d\bar{x} + C$$

Using $y(0) = 0$,

$$y = -\frac{U}{D}e^{-Kx^2/2D}\int\limits_0^x e^{\frac{K}{2D}\bar{x}^2}\, d\bar{x}$$

Using $y = \frac{U}{Kx}$, the characteristic length can be solved for using a series-expansion and a root-finding algorithm.

*Author contributions.* B.K. contributed to conceptualization, formal analysis, investigation, methodology, software, visualization, writing – original draft preparation, and writing - reviewing and editing. J.C. contributed to supervision and writing - reviewing and editing. T.H. contributed to supervision and writing - reviewing and editing. S.G. contributed to data curation and writing - reviewing and editing. S.J. contributed to conceptualization and supervision.



*Competing interests.* The authors declare no competing interests

*Acknowledgements.* The authors acknowledge the use of AI-based tools for language editing and coding assistance in the preparation of this manuscript. This research was conducted as part of an undergraduate thesis at Williams College.





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
