# Peer review of "A Fractal Framework for Channel-Hillslope Coupling"

_EGUsphere, 2024_

## Author Comment (AC1)

**1    Characteristic Length of Gabilan Mesa**

We thank David Litwin for his review, in which he raises an important point about the difference between the characteristic length of 62 m on Gabilan Mesa, derived from a slope-area analysis, and the much shorter characteristic length scale presented in Perron et al. (2009). In Perron et al. (2009), the authors perform a regression of $\frac{|\nabla z|}{\nabla^2 z - C_{ht}}$ against drainage area to estimate $\frac{D}{K}$ and $m$, assuming a stream-power formulation with $n = 1$ (Equation 1 in our work). This analysis does not account for grid resolution, which appears to be 5 meters.

In our work, we demonstrate that the drainage area on hillslopes is not independent of grid resolution. Therefore, the regression for $\frac{D}{K}$ and $m$ presented in Perron et al. (2009) is dependent on grid resolution and is not a valid method for extracting these parameters.

An improvement could involve using specific drainage area, $\frac{A}{\delta}$ for hillslopes and $\frac{A}{w}$ for channels. However, to validate this method for extracting $\frac{D}{K}$ and $m$, multiple tests should be conducted across a range of resolutions to ensure that there is no grid-resolution dependence. This is challenging, as grid resolution also affects slope and curvature calculations (Grieve et al., 2016b). Regressions based on slope and curvature often introduce substantial variability, even when the topography is smoothed, which reduces the reliability of these regressions.

While $r$ is indeed the characteristic length for linear diffusive topography with $n = 1$, as shown in our simulations, real-world conditions—such as non-uniform runoff and weakly nonlinear diffusion—often violate these assumptions. For slopes approaching the critical gradient in the Andrews-Bucknam/Roering model, the characteristic length to the inflection point will exceed $r$, as observed in a one-dimensional analysis. In this case, an additional dimensionless group, $\frac{U}{\sqrt{DKS_c}}$, modulates the extent to which the horizontal length scale is influenced by nonlinear diffusion. Although Gabilan Mesa serves as a useful case study for stream-power plus linear diffusion, it still shows some effects of nonlinear diffusion, even if they are minor, as shown by Grieve et al. (2016a). Given the consistency between our numerical simulations and Gabilan Mesa, we consider it likely that $r \approx 62m$ for Gabilan Mesa. However, as we have emphasized, extracting $K$ to validate this assumption is difficult, and we can only assume that our assumptions are reasonably met by referring to this value as $r$. We thank David Litwin once again for this comment and plan to discuss this issue in greater detail in the revised version.

**2    Contour Curvature**

Comment on line 93: Hillslopes do not just have parallel flowpaths, right?

We reference the concept of parallel flow paths on hillslopes as the intuition presented in previous research. On line 207, we address flow divergence ($0 < Df < 1$) and convergence ($1 < Df < 2$). This relationship is based on intuition, as we currently lack a rigorous proof connecting $D_f$ and contour curvature.

**References**

Grieve, S. W. D., Mudd, S. M., Hurst, M. D., and Milodowski, D. T.: A nondimensional framework for exploring the relief structure of landscapes, Earth Surface Dynamics, 4, 309–325, https://doi.org/10.5194/esurf-4-309-2016, 2016a.

Grieve, S. W. D., Mudd, S. M., Milodowski, D. T., Clubb, F. J., and Furbish, D. J.: How does grid-resolution modulate the topographic expression of geomorphic processes?, Earth Surface Dynamics, 4, 627–653, https://doi.org/10.5194/esurf-4-627-2016, 2016b.

Perron, J., Kirchner, J., and Dietrich, W.: Formation of evenly spaced ridges and valleys, Nature, 460, 502–505, https://doi.org/10.1038/nature08174, 2009.

---

## Author Response (AR1)

**Reply to Reviewers**
Ben Kargère et al.

We thank the reviewers for their constructive and thorough feedback on our manuscript. We appreciate that good reviews take time, and would like to thank the reviewers for taking the time to provide these extremely helpful reviews. Both reviewers highlighted technical areas that could be more fully explained and suggested areas where the research questions could be better clarified. In response to these major comments we have added a paragraph explaining our reasoning. We have clarified the text to address the technical points made and these are described as a response to each specific comment below. We have also attached a tracked changes manuscript.

We have carefully addressed the concerns raised and have made significant revisions to improve clarity and rigor. Below, we provide detailed responses to each point raised by the reviewers, highlighting the corresponding changes made to the manuscript.

**1 Reply to review by Tyler Doane**

**1.1 Main Comments**

**Comment:** "Landscapes have many fractal qualities and things like fractal dimensions are useful in many contexts. However, fractal definitions often break down at smaller scales because scaling relationships simply do not continue at smaller and smaller scales"
**Reply:** We appreciate the comment. We now make clear in the text that when analyzed at sufficiently high resolution, contributing drainage regions, which are two-dimensional for channels (hence drainage area), break down on hillslopes, giving rise to a system of contributing drainage dimension that does not obey self-similarity. We have added a sentence in the introduction to further clarify this point.
**Original:** ... instead varying with the grid resolution according to their fractal dimension.
**Revised:** ... instead varying with the grid resolution according to their fractal dimension. In other words, when analyzed at sufficiently high resolution, the scaling relationship for contributing drainage regions, which are two-dimensional for channels (hence drainage area), breaks down on hillslopes, resulting in a system of contributing drainage dimension that does not exhibit self-similarity.

**Comment:** "So, is there a lower limit for this analysis? What if we had a 1-cm grid resolution DEM? I suspect that there are contributing areas with fractal dimension of 2 that would be classified as "hillslope" in the field."
**Reply:** By definition, contributing drainage regions with a fractal dimension of two are independent of grid resolution. Microtopographic roughness on small scales influences nodes with a contributing drainage fractal dimension of less than two. This effect is evident in the flow-path striations shown in Fig. 8. The use of dimension two as a channel-head extraction method remains resolution independent, provided that $\delta << r$, so a 1cm resolution DEM would produce a channel extraction similar to a 1m or even a 10m DEM. We have added a sentence to further clarify this point in the Gabilan Mesa analysis section.
**Original:** The drainage area of channel heads is resolution-independent. Figure 8 confirms that this corresponds to the location of channel heads as identified by the contour curvature method of Pelletier (2013).
**Revised:** Both Fig. 7 and Fig. 8 demonstrate that the drainage area of channel heads is resolution-independent, as predicted by Fig. 5 and Eq. (12). Figure 8 confirms that the location of channel heads, nodes with $A = (62m)^2$ and thus $D_f = 2$ with $r = 62m$ in Eq. (12), matches the location of channel heads as identified by the contour curvature method of Pelletier (2013).

**Comment:** "It is not exactly clear to me how the non-dimensionalization presented earlier in the manuscript figures in to the later parts. If that connection can be made more explicitly, I think that the clever dimensional analysis that they did might have more impact. I note that r is consistent throughout, but to me, that seems to just be the critical contributing area."
**Reply:** Thanks for highlighting the need to make this connection more explicit. Non-dimensionalization in one dimension is not strictly necessary, as we could have presented a solution for $r$ using more conventional methods (detailed in the appendix), which may be less intuitive for readers. The non-dimensionalization of

the one-dimensional equation allows us to build intuition in the one-dimensional case while also connecting with the literature. The value $CI \approx 6.828$ sets the distance between the inflection point and the boundary length, $\ell$ (Fig. 1). Using the equation for CI, we can express the boundary length in terms of D and K, and therefore solve for the length from the topographic maximum to the inflection points given that it is one half of the boundary length. This is the solution presented in the paper. In addition to the revision below, we have added clarifying phrases to help connect the one-dimensional result and the fractal dimension definition in our revisions in response to other comments, in particular David Litwin's comment on line 56 of the original manuscript.

**Original:** $r$ can also be solved for from Eq. (2) as a first-order ordinary differential equation in $dz/dx$ (Appendix 1).

**Revised:** While $r$ can also be derived from Eq. (2) as a first-order linear ordinary differential equation in $dz/dx$ (Appendix 1), the insights gained from the non-dimensionalization of the one-dimensional equation are useful for analyzing the landscape evolution equation in two horizontal dimensions.

**1.2 Line Comments**

**Comment on line 79:** "The flux does not involve the absolute value of slope for linear diffusion. Flux is a vector and therefore the sign of the slope matters."

**Reply:** Thanks for catching this typo - we have corrected it and checked for any other instances of this error within the text.

**Original:** ...$q_d$, is directly proportional to the gradient of $z$, given as $q_d = D|\nabla z|$. The divergence of the diffusive flux, $D\nabla^2 z$ can be positive or negative...

**Revised:** ...$q_d$, is directly proportional to the gradient of $z$, given as $q_d = -D\nabla z$. The divergence of the diffusive flux, $-D\nabla^2 z$ can be positive or negative...
* * *
**Comment on line 87:** "U should have units $LT^{-1}$, not $HT^{-1}$"

**Reply:** U has fundamental dimension $HT^{-1}$, even though the height scale can be expressed with units of length, like meters, for example. We included an additional sentence to further highlight this concept.

**Original:** We define distinct height ($H$) and length ($L$) dimensions, in alignment with prior dimensional analyses (Theodoratos et al., 2018). For steady-state topography, the horizontal dimension pertains to the two-dimensional domain, while the vertical dimension serves as the function's codomain, organized so that erosion balances uplift everywhere.

**Revised:** We define distinct height ($H$) and length ($L$) dimensions, in alignment with prior dimensional analyses (Theodoratos et al., 2018; Willgoose et al., 1991). Although both $H$ and $L$ can be expressed in meters, treating them as separate dimensions facilitates our ability to simplify the model through non-dimensionalization (Huntley, 1967). For steady-state topography, the horizontal dimension pertains to the two-dimensional domain, while the vertical dimension serves as the function's codomain, organized so that erosion balances uplift everywhere.
* * *
**Comment on line 100:** "It seems like $\hat{t}$ has units $LT^{-1}$ so that $d\hat{z}/d\hat{t}$ has units of $TL^{-1}$. That would make equation (3) incorrect."

**Reply:** $\hat{t}$ is dimensionless, as are $d\hat{z}$ and $d\hat{z}/d\hat{t}$. We made a typo in this section, and it should read $\frac{t}{t_{\hat{c}}} = \hat{t}$. Equation 3 is correct and is dimensionless.

**Original:** $\frac{z}{t_{\hat{c}}} = \hat{t}$

**Revised:** $\frac{t}{t_{\hat{c}}} = \hat{t}$
* * *
**Comment on line 116:** Why are the authors using this expression for CI? Where does it come from?

**Reply:** We acknowledge the potential for confusion and have edited the text to clarify our approach. CI is the one free dimensionless group (a Péclet number) derived from this non-dimensionalization, given $m = 1$ (in one dimension) and $n = 1$. Since we have four parameters (U,D,K, and $\ell$), and three fundamental dimensions (H, L, T), we can derive a single dimensionless group.

**Original:** $C_I$, the channelization index, is a Péclet number, explaining the competition between advection and diffusion (Perron et al., 2008; Anand et al., 2023; Bonetti et al., 2020).

**Revised:** Given the four parameters ($U$, $D$, $K$, and $\ell$) and three fundamental dimensions ($H$, $L$, $T$) in Eq. (2), the equation can be rewritten using non-dimensionalization to include a single dimensionless group (Buckingham, 1914). This dimensionless group, referred to as the channelization index $C_I$, functions as a

Péclet number and quantifies the competition between advection and diffusion in the domain (Perron et al., 2008; Anand et al., 2023; Bonetti et al., 2020).
* * *
**Comment on line 136:** "The authors refer to multiple-flow direction and D-infinity. It seems like they mean D-infinity. They are different."

**Reply:** We acknowledge the potential for confusion here, but seek to clarify that we only referenced multiple-flow direction algorithms on line 165 of the original manuscript. We have edited this phrase to mention the D-infinity, instead, and hope this has clarified any previous ambiguities.

**Original:** Line 165 (of the original manuscript): The dimensionless group $\frac{r}{\delta}$ represents the number of pixels of contributing drainage, though not necessarily individual pixels (with multiple flow-direction routing algorithms), required to form an inflection point.

**Revised:** The dimensionless group $\frac{r}{\delta}$ represents the number of pixels of contributing drainage, though not necessarily individual pixels (with the $D\infty$ flow routing algorithm), required to form an inflection point.
* * *
**Comment on line 186:** "This paragraph is a bit tough to get through. Sentences like "Lines with no width, have width delta" are a bit challenging. Is there another way to say this that does not have apparently contradictory statements?"

**Reply:** We are grateful for the feedback and have edited the text accordingly.

**Original:** Lines, with no width, have width $\delta$.

**Revised:** On a two-dimensional mesh, lines are represented with a width equal to the grid resolution.
* * *
**Comment on Equation 11:** "Can this be explained a bit more? It seems like much of the take-away of this manuscript relies on readers understanding how the fractal dimension is calculated. It's not clear to me where this comes from."

**Reply:** We have expanded the explanation of equation 11 within the text in order to make things clearer for the reader.

**Original:** The numerator $\log(\frac{a}{\delta})$ is a number of contributing pixels. The denominator, $\log(\frac{r}{\delta})$ is the magnification factor, one of the dimensionless groups, referring to number of contributing pixels at the inflection point.

**Revised:** The numerator, $\log(\frac{a}{\delta})$, is the logarithm of the number of contributing pixels. The denominator, $\log(\frac{r}{\delta})$ is the logarithm of the magnification factor, the dimensionless group $\frac{r}{\delta}$, referring to number of contributing pixels at the inflection point. Therefore, for each node, Eq. (11) is calculated by comparing the logarithm of the number of contributing pixels to the logarithm of the number of pixels at the inflection point (Fig. 5).
* * *
**Comment on Figure 7:** "This is an interesting take on a classic plot! I think that it could be made a bit clearer. How does this differ from what one might infer from a classic approach to identifying channel-hillslope transition points?"

**Reply:** We appreciate the reviewer's support. Figure 8 illustrates the correspondence between our method and the Pelletier contour-curvature derived algorithm. However, plotting the drainage areas from these contour-curvature derived channel heads on Figure 7 is not beneficial. This is because the contour curvature method is based on the 1m DEM, so these channel heads are shifted across resolution by position only, without regard to the actual drainage areas. Therefore, any plotted drainage areas might simply reflect the effects of resampling rather than the effectiveness of the method itself. Despite this, the channel head locations do align in Figure 8, which can be seen in more detail in the below figure.

**Supplementary Figure:**

[Figure]

Figure 1: Zoomed in version of Fig. 8.

**Comment on Line 280:** "The same must be done for linear models, yes?"

**Reply:** We appreciate the reviewer's insightful comment. By this statement, we mean that nonlinear diffusion results in a nonlinear effect on grid resolution. Specifically, a smaller grid resolution can result in steeper local slopes, leading to increased flux. This change means that total flux is not conserved, as detailed in Ganti et al. (2012).

**Original:** Additionally, computational models incorporating nonlinear diffusion must address the effect of the grid resolution setting the spatial scale of the diffusion process (Ganti et al., 2012; Furbish and Roering, 2013).

**Revised:** Additionally, computational models incorporating nonlinear diffusion must address the nonlinear effects of the grid resolution setting the spatial scale of the diffusion process (Ganti et al., 2012; Furbish and Roering, 2013).

**2 Reply to review by David Litwin**

**2.1 Main Comments**

**Comment:** "First is that the paper needs overall revision to help the reader understand how the problem is related to the analyses conducted, and how the data comparison is related to the analyses."

**Also comment on line 225:** "Before talking about Gabilan Mesa, tell us what aspect of your results you are attempting to test or confirm, and how you will know if you were successful."

**Also comment on lines 235-236:** "The pieces are here, but as I mention before, we need a clearer indication of what you hypothesized and what you see. As I see it, your first hypothesis is that the transition from hillslope to colluvial is resolution-dependent and occurs at A = r*delta. Your second hypothesis is that channel head locations are resolution independent, and located at a distance related to the length r that you identified testing hypothesis 1, specifically A=r2. Optional, but if you wanted to add something for quantitative comparison to Figure 8, you could show a box and whisker plot of drainage area at the Pelletier channel heads for each resolution, with a horizontal line showing the length scale squared."

**Reply:** The feedback here is greatly appreciated, and we acknowledge the importance of clarifying our hypotheses. To that end, we have added to the introduction and confirmation sections. We have also included a sentence in the abstract to highlight the application of this theory to real-world landscapes, specifically in the case of Gabilan Mesa.

**Original (in the introduction):** We demonstrate that this corresponds to both computational simulations and real-world topography using Gabilan Mesa in California. Finally, we propose directions for computational models and suggest that real-world landscapes have a property analogous to a grid resolution.

**Revised (from original paragraph in introduction):**
We demonstrate that this theory, derived from one-dimensional analyses and computational simulations, corresponds to real-world topography using Gabilan Mesa in California. In particular, we show that on Gabilan Mesa, the drainage area of nodes with the steepest slope (the inflection point) scales with one factor of grid resolution. In contrast, the drainage area of channel heads, calculated as the square of the drainage area divided by the grid resolution at the inflection point, is independent of grid resolution. Using

these results, we propose directions for computational models and suggest that real-world landscapes have a property analogous to a grid resolution.

**Revised (new paragraph in Confirmation section):** To apply this theory to real-world landscapes suitable for the stream-power plus linear diffusion model, it is necessary to identify a set of hypotheses to test. Our first hypothesis is that the drainage area at the inflection point scales with one factor of grid resolution. Second, we hypothesize that the drainage area of channel heads is independent of grid resolution and corresponds to the square of the specific drainage area at the inflection point.

**Revised (new sentence in abstract):** We substantiate this theory with topographic analyses of Gabilan Mesa, California.
* * *
**Comment:** "Second, I think there is one important element missing to the discussion of characteristic scales at Gabilan Mesa. The authors estimate a characteristic length scale of 62 m, based on the rollover point in the slope-area relationship. They acknowledge that this may not be equal to the characteristic length scale, r, and explain that the parameter K in r is poorly constrained so they could not calculate it. Perron et al. (2009) got around this limitation with a topographic analysis that uses the whole streampower+diffusion model, and estimated the characteristic scale Lc=17.2 m at Gabilan Mesa. With the algebraic factor that relates Lc and r, r=22.5 m (I think), which is significantly smaller than that estimated from slope-area analysis. Using this value in the fractal analysis would give significantly different values of both the hillslope-colluvial and the colluvial-channel transition points. To my mind, your analysis and this discrepancy suggest that the rollover point in slope-area (at least at Gabilan Mesa) and r in the streampower+diffusion model are functionally similar scales, even if the values aren't close. This is despite Gabilan Mesa being a sort of type-case site for the streampower+diffusion model. This is not too far from what you have! But I think it is worth presenting the Perron et al. value and elaborating on the discrepancy but functional similarity of r (or Lc) and the inflection point as characteristic scales"

**Reply:** We have replied to this comment in detail in a previous discussion reply, echoing the ideas presented in the revised version.

**Discussion Reply:** "We thank David Litwin for his review, in which he raises an important point about the difference between the characteristic length of 62 m on Gabilan Mesa, derived from a slope-area analysis, and the much shorter characteristic length scale presented in Perron et al. (2009). In Perron et al. (2009), the authors perform a regression of $\frac{|\nabla z|}{\nabla^2 z - C_{ht}}$ against drainage area to estimate $\frac{D}{K}$ and $m$, assuming a stream-power formulation with $n = 1$ (Equation 1 in our work). This analysis does not account for grid resolution, which appears to be 5 meters.

In our work, we demonstrate that the drainage area on hillslopes is not independent of grid resolution. Therefore, the regression for $\frac{D}{K}$ and $m$ presented in Perron et al. (2009) is dependent on grid resolution and is not a valid method for extracting these parameters.

An improvement could involve using specific drainage area, $\frac{A}{\delta}$ for hillslopes and $\frac{A}{w}$ for channels. However, to validate this method for extracting $\frac{D}{K}$ and $m$, multiple tests should be conducted across a range of resolutions to ensure that there is no grid-resolution dependence. This is challenging, as grid resolution also affects slope and curvature calculations (Grieve et al., 2016c). Regressions based on slope and curvature often introduce substantial variability, even when the topography is smoothed, which reduces the reliability of these regressions.

While $r$ is indeed the characteristic length for linear diffusive topography with $n = 1$, as shown in our simulations, real-world conditions—such as non-uniform runoff and weakly nonlinear diffusion—often violate these assumptions. For slopes approaching the critical gradient in the Andrews-Bucknam/Roering model, the characteristic length to the inflection point will exceed $r$, as observed in a one-dimensional analysis. In this case, an additional dimensionless group, $\frac{U}{\sqrt{DKS_c}}$, modulates the extent to which the horizontal length scale is influenced by nonlinear diffusion. Although Gabilan Mesa serves as a useful case study for stream-power plus linear diffusion, it still shows some effects of nonlinear diffusion, even if they are minor, as shown by Grieve et al. (2016a). Given the consistency between our numerical simulations and Gabilan Mesa, we consider it likely that $r \approx 62m$ for Gabilan Mesa. However, as we have emphasized, extracting $K$ to validate this assumption is difficult, and we can only assume that our assumptions are reasonably met by referring to this value as $r$."

**Original:** Figure 7 plots the gradient by drainage area for the section of the DEM shown in Fig. 8. The characteristic length corresponding to the inflection point is 62m, the drainage area of which varies by a factor

of $\delta$ across resolutions (Bernard et al., 2022). The drainage area of channel heads is resolution-independent. Figure 8 confirms that this corresponds to the location of channel heads as identified by the contour curvature method of Pelletier (2013). While $D$ can be consistently defined by hilltop curvature, $K$ is a relatively free parameter. There are several approaches for determining $K$, such as by drainage density or by channel steepness as $k_{sn} = (U/K)^{-m/n}$. However, this would assume that erodibility is constant throughout the domain (both hillslopes and channel networks). For this reason, as well as the potential for weakly nonlinear diffusion, we refrain from defining the characteristic length as $r$ in this case-study. Nonetheless, the coherence between the topographic analyses presented in Fig. 7 and Fig. 8 and the mathematical framework derived from computational results confirms the utility of our theory (Bras et al., 2003).

**Revised:**

Figure 7 plots the gradient by drainage area for the section of the DEM shown in Fig. 8. The characteristic length corresponding to the inflection point is 62m, the drainage area of which varies by a factor of $\delta$ across resolutions (Bernard et al., 2022). The drainage area of channel heads is resolution-independent. Figure 8 confirms that this corresponds to the location of channel heads as identified by the contour curvature method of Pelletier (2013).

While $D$ can be consistently defined by hilltop curvature, $K$ is difficult to estimate from topography. Potential approaches include using drainage density (Tucker and Bras, 1998) or channel steepness (Whipple and Tucker, 1999). These approaches are problematic, since identifying $K$ from channel steepness or drainage density would assume that erodibility is constant throughout the domain (both hillslopes and channel networks), which is unlikely given variations in sediment cover and physical processes. Another approach is outlined by Perron et al. (2009), who conducted a regression of $\frac{|\nabla z|}{\nabla^2 z - C_{ht}}$, where $C_{ht}$ is the hilltop curvature, against drainage area to derive values for $\frac{D}{K}$ and $m$, assuming a stream-power formulation with $n = 1$ (Eq. 2). However, this analysis overlooks grid resolution, and as demonstrated in Fig. 7, drainage areas on hillslopes are not independent of grid resolution. Consequently, this regression is not a valid method for extracting $m$ and $\frac{D}{K}$.

Given the consistency between our numerical simulations and the topography of Gabilan Mesa, we consider it likely that $r = 1.3\sqrt{\frac{D}{K}} \approx 62m$ for this case-study, though we leave rigorous validation of these parameters, $K$ in particular, for future work. Nonetheless, the coherence between the topographic analyses presented in Fig. 7 and Fig. 8 and the mathematical framework derived from computational results confirms the utility of our theory (Bras et al., 2003).

**2.2 Line Comments**

**Comment on lines 16-19:** "The end of the abstract could use some more thought. It's not clear what "observed scaling of channel width" is being referred to, nor why it is logical that "real-world landscapes have something analogous to the concept of grid resolution.""

**Reply:** We are grateful for the feedback and have edited the text accordingly.

**Original:** This conceptualization aligns with the observed scaling of channel width. It also importantly suggests that real-world landscapes have something analogous to the concept of a grid resolution, as this paper demonstrates. In doing so, our works clarifies several unresolved properties of channel-hillslope coupling, with potential for substantially improving the accuracy of coupled landscape evolution models in replicating landscape forms.

**Revised:** This conceptualization aligns with the scaling of channel width as the square root of drainage area. Since channel heads form at a resolution-independent drainage area, the width of channel heads is not explicitly defined, suggesting that the grid resolution is analogous to the property of channel head width in real-world landscapes, influenced by the particle size. We validate this theory with topographic analyses of Gabilan Mesa, California. These findings clarify several unresolved properties of channel-hillslope coupling, with potential for substantially improving the accuracy of coupled landscape evolution models in replicating landscape forms.

**Comment on lines 26-28:** "I'm not quite sure what this sentence is trying to say."

**Reply:** We appreciate the need to clarify and have edited the text to improve intelligibility.

**Original:** In real-world landscapes, this transition is often described in terms of stochastic perturbations and time-dependent behavior (Smith and Bretherton, 1972; Howard and Kerby, 1983; Del Vecchio et al., 2023).

Given the prohibitive timescales of landscape evolution, the computational evaluation of LEMs is essential for elucidating landscape evolution over extended periods, further providing insights into the interconnected dynamics of hillslope and channel evolution (Tucker and Hancock, 2010).

**Revised:** In real-world landscapes, this transition is often described in terms of stochastic perturbations and time-dependent behavior (Smith and Bretherton, 1972; Howard and Kerby, 1983; Del Vecchio et al., 2023). Given that landscape evolution occurs over long time scales, computationally evaluated LEMs are commonly used to test theories describing channel initiation (Anand et al., 2022; Tucker and Hancock, 2010).
* * *
**Comment on line 38:** "When you say a threshold for 'slope-area', are you referring to thresholds for streampower, and other terms with the same dimensions as $A^m * S^n$ (e.g., Theodoratos and Kirchner, 2020)?"

**Reply:** We appreciate the need for clarification and have accordingly amended the text.

**Original:** Instead, computational models often implement a physically-derived or arbitrarily chosen threshold for drainage area or slope-area, below which stream-power erosion is absent (Perron et al., 2008; Tucker and Bras, 1998; Campforts et al., 2017).

**Revised:** Instead, computational models often implement a physically-derived or arbitrarily chosen threshold for drainage area or the product of powers of drainage area and slope ($A^m S^n$), below which stream-power erosion is absent (Perron et al., 2008; Tucker and Bras, 1998; Campforts et al., 2017; Theodoratos and Kirchner, 2020).
* * *
**Comment on line 56:** "I think the connection between what you have been discussing what you say you will do could be clearer. What is the specific value of the 1D analysis? How does the box counting method and fractal dimension build on or relate to your 1D analysis?"

**Reply:** The revised version below emphasizes the link between the one-dimensional analysis and our box-counting fractal definition. We think that explaining the box-counting method and fractal dimension in more detail in the introduction could potentially confuse the reader, particularly because of the absence of visual aids such as Figure 5.

**Original:** In this work we analytically derive the characteristic landscape length from a one-dimensional analysis. We define a fractal box-counting definition using the characteristic landscape length and the pixel width as a measure. Unchannelized nodes, those with contributing drainage less than the characteristic length squared, do not have a well-defined contributing drainage area, instead varying with the grid resolution according to their fractal dimension.

**Revised:** In this work we analytically derive the characteristic landscape length, defined as the contributing length to an inflection point in a one-dimensional analysis. On a two-dimensional domain, we use the characteristic landscape length and the pixel width as a measure to define a fractal box-counting definition. This reveals that unchannelized nodes, those with contributing drainage area less than the characteristic length squared, do not have a well-defined contributing drainage area, instead varying with the grid resolution according to their fractal dimension.
* * *
**Comment on line 79:** "Check signs. Diffusion processes have flux inversely proportional to gradient (from Fick's Law). The positive sign on the diffusion term in (1) comes from something like: dz/dt = U – Ef – Eh, Eh = -D nabla$^2 z$"

**Reply:** This comment is addressed in our response to Tyler Doane's comment on the same line.
* * *
**Comment on line 93:** "Hillslopes do not just have parallel flowpaths, right? Divergence of flow paths is also an important part (Bogaart and Troch, 2006)."

**Reply:** We reference the concept of parallel flow paths on hillslopes only when discussing previous research. On line 207, we address flow divergence ($0 < D_f < 1$) and convergence ($1 < D_f < 2$). This relationship is based on intuition, as we currently lack a rigorous proof connecting $D_f$ to plan curvature. The revised version below stresses that the general concept originates from previous work from which we are building on.

**Original:** As noted by Pelletier (2010), Hergarten (2020), and Hergarten and Pietrek (2023), hillslopes are thought to have parallel flow paths, whereas channels have convergent flow paths. On hillslopes, the width of the parallel flow paths is a function of grid resolution, whereas regions with convergent flow are relatively unaffected by changes in grid resolution.

**Revised:** As noted by Pelletier (2010), Hergarten (2020), and Hergarten and Pietrek (2023), hillslopes are thought to have parallel flow paths, whereas channels have convergent flow paths. As these authors suggest,

the area of contributing drainage regions with parallel flow paths is determined by the grid resolution, which dictates their width. In contrast, regions with convergent flow have contributing drainage areas that are relatively unaffected by changes in grid resolution.
* * *
**Comment on line 106** "I think '3' might be out of place."

**Reply:** We appreciate the correction and have amended the manuscript by removing this out of place reference to Fig. 3.

**Original:** Larger values of $C_I$ manifest in narrower hillslope profiles relative to the boundary size 3, formed by the relative strength of advective processes over diffusive processes.

**Revised:** Larger values of $C_I$ manifest in narrower hillslope profiles relative to the boundary size, formed by the relative strength of advective processes over diffusive processes.
* * *
**Comment on lines 139-143** "You've already discussed the choice of m=1/2, n=1, so I would lead with (142) – the topic here is about the scaling of channel width."

**Reply:** While we appreciate the reviewer's perspective and insight, we respectfully disagree on this point. In addition to the reasons listed in the text, the choice m=$\frac{1}{2}$, n=1, without addressing the grid resolution, is problematic, since it (1) only applies to channels, and (2) is based on $w \propto \sqrt{A}$, without actually including the grid resolution (the coefficient in the width power-law) in K, which then changes the value of K.

**Original:**

**2.2.1 First Section**

[revised manuscript text omitted]

**2.2.6  3rd Section**

In Eq. (2), $D$ has fundamental dimension $L^2T^{-1}$, and $U$ has fundamental dimension $HT^{-1}$. Throughout this work we specify $K$ as having the fundamental dimension $T^{-1}$.
* * *
**Comment on line 145.** "Citation style."
**Original:** Channel widths are observed to scale with the square root of drainage area Leopold and Maddock Jr (1953), thus $\frac{w}{\delta} \propto \frac{\sqrt{A}}{\delta}$.
**Revised:** Channel widths are observed to scale with the square root of drainage area (Leopold and Maddock Jr, 1953), thus $\frac{w}{\delta} \propto \frac{\sqrt{A}}{\delta}$.
* * *
**Comment on line 150:** "This is slightly pedantic, but this is assuming uniform generation of runoff, rather than just uniform precipitation."
**Reply:** We are grateful for the feedback and have edited the text accordingly.

**Original:** This normalization implies that fluvial erosion is proportional to contributing area, and thus discharge, assuming a uniform precipitation rate.

**Revised:** This normalization implies that fluvial erosion is proportional to contributing area, and thus discharge, assuming a uniform runoff generation rate (Litwin et al., 2022).
* * *
**Comment on line 152:** "the contributing drainage of parallel locally parallel"

**Original:** The use of $a$ ensures that the contributing drainage of parallel locally parallel on hillslopes are independent of pixel width, thereby conforming to a one-dimensional framework. We will show that parallel flows occur locally for at the inflection point, $a = r$.

**Revised:** The use of $a$ ensures that the contributing drainage of locally parallel flow regions on hillslopes are independent of the grid resolution, thereby conforming to a one-dimensional framework. We will show that locally parallel flows occur at the inflection point, $a = r$.
* * *
**Comment on lines 165-166:** "You could note this quantity (Pi1) is held constant by Theodoratos et al. (2018) and other papers."

**Reply:** We are grateful for this suggestion and have edited the text accordingly.

**Original:** The dimensionless group $\frac{r}{\delta}$ represents the number of pixels of contributing drainage, though not necessarily individual pixels (with the $D\infty$ flow routing algorithm), required to form an inflection point.

**Revised:** The dimensionless group $\frac{r}{\delta}$ represents the number of pixels of contributing drainage, though not necessarily individual pixels (with multiple flow-direction routing algorithms), required to form an inflection point. In the literature, some papers keep this value constant (Theodoratos et al., 2018), but others do not (Anand et al., 2023).
* * *
**Comment on lines 176-179:** "You might want to be clear that this is an inflection point in topography, but a maximum in slope-area space (as shown in Figure 2). More generally, I'm having a hard time seeing what you describe in Figure 2. Maybe put some scaling lines to help guide the reader. Smaller points, and a binned trend like in Figure 7 could help."

**Reply:** We appreciate the suggestions and have have edited the figure accordingly. In the revised version below, the dots are smaller, with an added binned median line in magenta. We added a vertical line at $a = \delta$ and labels for the specific drainage area values of the vertical lines in order to make this plot more similar to Fig. 7. We added the local scaling regions at the channel heads and for the channels, as suggested. To further improve the visibility and to simplify the plot, we substituted the original legend for the labels of the local scaling regions. We also bracketed the exponent of year in our U value so that it reads $yr^{-1}$ and not $yr^-1$, in both this figure and in Fig. 3.

**Revised:**

[Figure]

Figure 2: Slope-specific drainage area plot for the simulated topography shown in Fig. 3 for $\ell = 1000$, $\delta$ $= 10$, $K = 2.5e - 5yr^{-1}$, $D = 7e - 2m^2yr^{-1}$, $U = 3.6e - 3myr^{-1}$. The binned median slope is plotted in magenta. The specific drainage area $a = r$, shown with the red line, has the steepest slope. The specific drainage area value $\frac{r^2}{\delta}$, shown in orange, represents the transition to fluvial power-law scaling, with slope decreasing steadily with contributing area. This value is resolution independent as an area, but not as a specific drainage area, which has the fundamental dimension of length.
* * *
**Comment on Figure 2:** "'This value is resolution independent as an area, but not as a length.' It's not clear to me what you mean by this."
**Reply:** We acknowledge the potential for confusion and have edited the text to clarify this sentence.
**Original:** This value is resolution independent as an area, but not as a length.
**Revised:** This value is resolution independent as an area, but not as a specific drainage area, which has the fundamental dimension of length.
* * *
**Comment on Figure 3:** "Last sentence of the caption is not clear. The yellow dots show $A \geq r^2$ rather than $A = r^2$."
**Original:** Channels form for $A = r^2$, shown with yellow dots.
**Revised:** Channels are highlighted with yellow dots for $A \geq r^2$.
* * *
**Comment on line 229:** "Citation style."
**Reply:** We also removed the reference to LSDTopotools in the following sentence, since we already mentioned that the data were from Grieve et al. (2016b).
**Original:** Our data originate from a 2015 DEM, as utilized by (Grieve et al., 2016b). These data include channel heads using the Pelletier algorithm (Pelletier, 2013), as implemented by the LSDTopoTools (Pelletier, 2013), which estimates channel head locations from contour curvature.
**Revised:** Our data originate from a 2015 DEM, as utilized by Grieve et al. (2016b). These data include channel heads using the Pelletier algorithm (Pelletier, 2013), which estimates channel head locations from contour curvature.
* * *
**Comment on Figure 7:** "Need to find a way to clean up this figure. The dots are too big and too dark, especially relative to the thin binned median line. The caption also needs to say that these are data from

Gabilan Mesa since this is the first figure with data rather than model output. At the moment, it just refers to following figure."

**Reply:** We appreciate the suggestions and have have edited the figure accordingly. To improve readability, we've reduced the dot size and removed the fractal dimension color mapping. The binned median line has been made thicker to enhance visibility. The caption mentions that these data are from Gabilan Mesa.

**Revised:**

[Figure]

Figure 3: Log-log plot of slope by drainage area for Gabilan Mesa, resampled across 1, 5, and 10 meter pixels, as shown in Fig. 8. The binned median is plotted in a magenta dashed line. The characteristic length is 62m. The minimum drainage area, corresponding to topographic maxima is $\delta^2$. The drainage area of the inflection point is $62\delta m^2$. Channel heads have the characteristic area of contributing drainage area which is resolution-independent.
* * *
**Comment on line 238:** "Estimating K this way also neglects the effect of hillslopes on channel steepness, both in the model (which is limited by assumptions made when coupling detachment-limited fluvial erosion and hillslope diffusion) and in real life. If you are interested, we also have a preprint in ESurf on this (Litwin et al., 2024). There is a way around this though, as shown in Perron et al. (2009). They did estimate something like r (their Lc), and found it to be much shorter, 17.2 m. See comments at the top about this."

**Reply:** While we appreciate the reviewer's perspective and insight, we respectfully disagree on the details in this point. The effect of hillslopes on channel steepness in SPD models (as discussed in Litwin et al. (2024)) is primarily due to the use of stream-power incision, rather than specific stream power incision. Considering that $K$ for rivers includes a factor $k_w$, which we show has the form $\frac{\delta}{r}$, the value of $K$ applied to rivers takes into consideration the effect of hillslopes on channel steepness, since $k_w$ includes both $\delta$, analogous to grain size, and $D$. Considering real-world implications, we address these in our revision in response to a main comment (below). We summarize these as "variations in sediment cover and physical processes." Concerning the second half of this comment, we explain why the regression on drainage area Perron et al. (2009) is inappropriate for hillslopes, since hillslopes have drainage area that is dependent on grid resolution, in our reply to one of the reviewer's main comments.

**Revised Section:** These approaches are problematic, since identifying $K$ from channel steepness or drainage density would assume that erodibility is constant throughout the domain (both hillslopes and channel networks), which is unlikely given variations in sediment cover and physical processes.
* * *
**Comment on lines 245-246:** "Again, I think it would be helpful to have scaling lines on one of these plots so we can see what m/n=1 and m/n=1/2 look like."

**Reply:** We address this in our response to the comment on lines 176-179.
* * *
**Comment on lines 248:** "The 'form A/w for channels' is vague."

**Reply:** We appreciate the need for clarification and have accordingly amended the text.

**Original:** This results in the form $\frac{A}{w}$ for channels, producing the correct scaling for channel slope but forgoing the spatial representation of widening channels.

**Revised:** This results in the form $\frac{A}{w}$, where $w$ is the channel width (Eq. 14), such that channel slope scales with drainage area according to the exponent $m = 1 - \alpha$, where $\alpha$ is the scaling of channel width (Pelletier, 2010). For self-similarity given Eq. (14), $\alpha = \frac{1}{2}$, and thus channels scale with drainage area according to $m = \frac{1}{2}$. However, applying $\frac{A}{w}$ to a single pixel forgoes the spatial representation of widening channels.
* * *
**Comment on lines 250-251:** "Interesting to see the argument for grid cell width as channel head width here! We argued for more or less the same in Litwin et al. (2022), section 7.4, but in that case it was on the basis of subsurface water transport capacity that affects the point at which channel heads emerge from saturation excess overland flow."

**Reply:** We have added a citation to Litwin et al. (2022).

**Original:** Rather than assuming that pixel widths exceed channel widths throughout the domain, we propose instead that models should assume that channel widths exceed the pixel width, with the pixel width equal to the channel head width.

**Revised:** Rather than assuming that pixel widths exceed channel widths throughout the domain, we propose instead that models should assume that channel widths exceed the pixel width, with the pixel width equal to the channel head width, as suggested by Litwin et al. (2022).
* * *
**Comment on line 267:** "Have shown that showed"

**Original:** Studies have shown that showed the particle size distribution is closely related to the transition from slope-invariant colluvial valleys (i.e. $1 < D_f < 2$) and fluvial channels, suggesting a maximum particle size at the channel head (Neely and DiBiase, 2023).

**Revised:** Studies have shown that the particle size distribution is closely related to the transition from slope-invariant colluvial valleys (i.e. $1 < D_f < 2$) and fluvial channels, suggesting a maximum particle size at the channel head (Neely and DiBiase, 2023).
* * *
**3 Correction to Figure 4**

In the caption of figure 4 we made a typo in one of our parameters. K was listed as $2.5e{-}4yr^{-1}$, while in fact for this model run the $K$ value was $5e{-}4yr^{-1}$. This error was also perpetuated in the channel highlighting, though with only a minimal effect. We reran this numerical simulation in order to verify and correct this error, and we included this reproduced model output in the revised version. We also checked that the parameters in the other figures matched those listed and we found no further errors.

**4 Typos**

**Reasoning:** We have fixed a typo in the Fig. 1 caption, which originally stated "asymptotically large values of $C_I$" twice in the same sentence.

[revised manuscript text omitted]